# Automatic Assessment of the 2-Minute Walk Distance for Remote Monitoring of People with Multiple Sclerosis

**DOI:** 10.3390/s23136017

**Published:** 2023-06-29

**Authors:** Spyridon Kontaxis, Estela Laporta, Esther Garcia, Matteo Martinis, Letizia Leocani, Lucia Roselli, Mathias Due Buron, Ana Isabel Guerrero, Ana Zabala, Nicholas Cummins, Srinivasan Vairavan, Matthew Hotopf, Richard J. B. Dobson, Vaibhav A. Narayan, Maria Libera La Porta, Gloria Dalla Costa, Melinda Magyari, Per Soelberg Sørensen, Carlos Nos, Raquel Bailon, Giancarlo Comi

**Affiliations:** 1Laboratory of Biomedical Signal Interpretation and Computational Simulation (BSICoS), University of Zaragoza, 50018 Zaragoza, Spain; 2Biomedical Research Networking Center in Bioengineering, Biomaterials and Nanomedicine (CIBER-BBN), 28006 Barcelona, Spain; 3Department of Microelectronics and Electronic Systems, Autonomous University of Barcelona, 08193 Bellaterra, Spain; 4Department of Medicine and Surgery, University Vita-Salute and Hospital San Raffaele, 20132 Milan, Italy; 5Danish Multiple Sclerosis Center, Department of Neurology, Copenhagen University Hospital Rigshospitalet, 2100 Copenhagen, Denmark; 6Multiple Sclerosis Center of Catalonia (CEMCAT), Department of Neurology/Neuroimmunology, Hospital Universitari Vall d’Hebron, Universitat Autonoma de Barcelona, 08035 Barcelona, Spain; 7Department of Biostatistics and Health Informatics, Institute of Psychiatry, Psychology and Neuroscience, King’s College London, London SE5 8AF, UK; 8Janssen Research and Development, LLC, Titusville, NJ 08560, USA; 9Institute of Psychiatry, Psychology and Neuroscience, King’s College London, London SE5 8AF, UK; 10Institute of Health Informatics, University College London, London NW1 2DA, UK; 11Davos Alzheimer’s Collaborative, Wayne, PA 19087, USA; 12Casa di Cura del Policlinico, 20144 Milan, Italy

**Keywords:** wearable device, accelerometer sensor, walk tests, disability level, fatigue severity

## Abstract

The aim of this study was to investigate the feasibility of automatically assessing the 2-Minute Walk Distance (2MWD) for monitoring people with multiple sclerosis (pwMS). For 154 pwMS, MS-related clinical outcomes as well as the 2MWDs as evaluated by clinicians and derived from accelerometer data were collected from a total of 323 periodic clinical visits. Accelerometer data from a wearable device during 100 home-based 2MWD assessments were also acquired. The error in estimating the 2MWD was validated for walk tests performed at hospital, and then the correlation (r) between clinical outcomes and home-based 2MWD assessments was evaluated. Robust performance in estimating the 2MWD from the wearable device was obtained, yielding an error of less than 10% in about two-thirds of clinical visits. Correlation analysis showed that there is a strong association between the actual and the estimated 2MWD obtained either at hospital (r = 0.71) or at home (r = 0.58). Furthermore, the estimated 2MWD exhibits moderate-to-strong correlation with various MS-related clinical outcomes, including disability and fatigue severity scores. Automatic assessment of the 2MWD in pwMS is feasible with the usage of a consumer-friendly wearable device in clinical and non-clinical settings. Wearable devices can also enhance the assessment of MS-related clinical outcomes.

## 1. Introduction

Multiple sclerosis (MS) is one of the most prevalent neurologic disorders, affecting over 2 million individuals worldwide [1]. Most people with MS (pwMS) experience their first symptoms between the ages of 20 and 40 [2], with gait being perceived among the most important domains [3] and fatigue among the most common symptoms [4]. Such symptoms have detrimental effects on physical ability, reducing, as a result, patients’ quality of life [5,6]. Although MS-related symptoms are unpredictable and vary in type and severity from one person to another, assessing walking limitations is a key factor to determine the degree and progression of clinical outcomes in MS [7].

Quantitative measurement of walking performance in individuals with MS is currently conducted in a clinical setting [8]. Walking capacity is assessed either by short distance tests, such as the Timed 25-Foot Walk (T25FW), or with middle distance tests, such as the 6-Minute Walk Test (6MWT) and its shorter version, called the 2-Minute Walk Test (2MWT). The T25FW assesses walking speed on a 25-foot linear path, and it is of high practical utility in the clinical setting due to the minimum requirement of time and space [9]. Walk endurance tests record the distance walked in 2 min (2MWD) or 6 min (6MWD) and provide information on motor fatigue not captured by short distance tests [7].

Several qualitative and semi-quantitative scales have been proposed to evaluate clinical outcomes in MS [10]. Among them, the Expanded Disability Status Scale (EDSS) and the Fatigue Severity Scale (FSS) are the most widely accepted measures of clinical disability [11] and fatigue [12], respectively. The EDSS evaluates the degree of disability based on several functional systems and ambulation, while the FSS quantifies fatigue severity based on cognitive and physical aspects of the individual. Previous studies have shown that the EDSS exhibits moderate-to-strong association with walk test scores [13,14,15], while contradictory results have been reported about the relation between walking endurance and fatigue severity scores [16,17].

Assessing walking performance in pwMS may improve sensitivity in detecting changes in objective and/or subjective clinical markers. However, since clinicians are actively involved, the evaluation of outcomes in clinical settings is expensive in terms of time and money. Furthermore, outcome measures that are assessed only at limited time-points might be insufficient for analyzing the disease progression over time. Therefore, automated estimation of walking test distance could be used as a tool for the early detection of disease progression. Screening tools offer the possibility to prevent burdens, decrease levels of unemployment, and reduce healthcare care cost escalation as MS symptoms gradually worsen [18].

Over the last few years, due to the ability of accelerometers to monitor physical activity, wearable devices have received great interest from the clinical perspective [19]. Gross motor activity, footsteps, and distance walked are some of the most common actigraphy-related parameters that have been used to assess walking behavior [20]. Numerous studies have demonstrated that wearable devices can be used for gait analysis or step detection, making them a practical tool for automating walk tests [21]. Recent studies have revealed that the MS population is the primary target for evaluating 6MWT-related measures through step counting and gait variability analyses [22,23]. Within these studies, the majority have been conducted in a stable environment, such as a research laboratory or clinical setting, and at a single time-point [24]. To the best of authors’ knowledge, only [25] investigated the association between clinical outcomes and gait-related features from 2MWTs performed out-of-clinic and daily over a 24-week period. However, the outcome of walking distance was estimated, and no validation with walk tests performed at clinical visits was conducted.

The aim of this study is to investigate the feasibility and utility of automatically assessing 2MWTs performed outside clinical settings for MS monitoring. First, the algorithm is validated for walk tests performed in clinical settings, and then, its performance is evaluated for walk tests performed at home. In addition to more frequent evaluation of walking performance, wearable devices can enhance the assessment of various clinical outcomes. Therefore, the association between the estimated distance and clinical outcomes, including 2MWD, 6MWD, T25FW, EDSS, and FSS, is investigated as well.

## 2. Materials and Methods

### 2.1. Dataset and Study Design

The current analysis was conducted in the context of the IMI2 Remote Assessment of Disease and Relapse—Central Nervous System (RADAR-CNS) program (https://www.radar-cns.org/, accessed on 1 April 2023). The aim of RADAR-CNS is to evaluate remote monitoring technologies in the context of CNS diseases. The MS branch of the project has recruited participants since 2018 across three European sites: Ospedale San Raffaele in Milan; Danish Multiple Sclerosis Center, Copenhagen University Hospital Rigshospitalet in Copenhagen; and Vall d’Hebron Research Institute in Barcelona. The RADAR-CNS protocol was designed in collaboration with a patient advisory board that provided feedback on a variety of user-facing aspects of the study, including the selection and frequency of survey measures, the functionality of the study app, participant-facing reports, the choice of the best participation incentives, the choice and use of wearable technology, and the data analysis strategy. Active data, such as walk distances and rating scales, were collected every three months at hospital. At each clinical visit, participants performed the 2MWT along a 10 m long corridor at the maximum possible speed. Then, during the following 7 days, they were instructed to perform the 2MWT at home, replicating the clinical setup. Participants were requested to wear the wearable device Bittium Faros (https://www.bittium.com/medical/bittium-faros, accessed on 1 April 2023) while performing the 2MWT both at hospital and at home. The Bittium Faros is a patch device that is placed at the chest and enables the recording of 3-axis accelerometer data. Bittium Faros is a certified medical device that can be used for ambulatory (Holter) recordings. In comparison to accelerometer data from smartphones or smartwatches, Bittium Faros offer a higher sampling frequency of up to 25 Hz (vs. 5 Hz).

The protocol was approved by the hospitals’ ethical committees, and all subjects provided informed consent. The inclusion criteria for this study were an age of 18 or older, a diagnosis of MS according to the 2010 changes to the McDonald criteria, Relapsing–Remitting Multiple Sclerosis (RRMS) or Secondary Progressive Multiple Sclerosis (SPMS) phenotypes, EDSS from 2 to 6, capacity to provide informed permission for participation, desire and ability to complete self-reported evaluations through smartphone, and possession of an Android smartphone. The existence of any condition (physical, mental, or social) that was likely to impair the subject’s capacity to comply with the protocol as well as pregnancy in female participants were exclusion criteria. The interested reader is directed to find for further information on the MS study and evaluation processes in [26].

In this study, data from 154 pwMS were collected from a total of 323 clinical visits. Demographics and clinical outcomes of participants are shown in Table 1. In addition to the 2MWD, clinical parameters such as the 6MWD, TF25FW, EDSS, and FSS were also available in most clinical visits. Around half of participants performed at least one 2MWT at home, resulting in 100 home-based possible assessments of the 2MWD.

### 2.2. Estimation of the 2MWD from Accelerometer Signals

A change in 2MWD could be due to a change in step length and/or a change in cadence. Many wearable technologies accurately estimate the number of steps, but step length is not usually estimated. Therefore, it makes sense to first study a fixed step length to evaluate if the number of steps alone can provide sufficiently accurate 2MWD estimates. Then, a second approach for assessing 2MWD based on estimation of participants’ step length is considered. First, the magnitude of the accelerometer was computed using the norm of the 3 axes. Then, the average magnitude was subtracted and a moving average filter (sliding window of length 0.3 s) was used to suppress noise and extreme values. Step identification was carried out by detecting local maxima that were greater than the standard deviation of the smoothed magnitude and separated by more than a minimum peak distance of 0.3 s. Finally, the distance was obtained by multiplying the total number of steps (*N*) with the step length (in meters).

In the first approach, a fixed step length (*L*) for each participant was established. In a large cross-sectional study, spatial and temporal parameters of gait in pwMS were investigated [27]. Participants were instructed to walk at a self-selected speed and at a fast speed. Patients were divided into levels of disability based on EDSS. Taking into account the fast speed walking and the mean disability level of the MS population used in this study (EDSS = 3.2, see Table 1), a step length L=0.724 m was chosen (EDSS 3.0–3.5 [27]). In the second approach, an individual-specific step length ($L_{i}$) was used. The average step length of an individual was obtained by combining the fixed length of the corridor and the number of steps in each lap ($N_{l}$). A moving standard deviation filter (sliding window of length 1 s) was applied to the magnitude of the accelerometer signal, and time instants with reduced acceleration due to the turning of an individual were identified (minimum peak distance of 4 s). Examples of the 2MWD estimation with the fixed step length, denoted D=N×L, and with the individual-specific step length, denoted $D_{i}=N\times L_{i}$, are illustrated in Figure 1.

### 2.3. Performance Evaluation and Correlation Analysis with Clinical Outcomes

To evaluate performance, the relative absolute error (RAE, ε) between the estimated distance at hospital, either with *D* or $D_{i}$, and the reference 2MWD was considered. It should be noted that the individual-specific step length ($L_{i}$) was estimated at the first clinical visit and was used for the rest of the 2MWTs performed either at hospital or at home. Statistical analyses were conducted to evaluate the performance between distance estimation methods using personalized and fixed step lengths. First, Pearson’s correlation coefficients (*r*) between the estimated distance (*D* or $D_{i}$) and the ground truth (2MWD) were estimated. Then, a z-test on Fisher z-transformed correlation coefficients was conducted. Moreover, the statistical difference of the error in estimating 2MWD with *D* and $D_{i}$ was examined. Two-sample F-tests to ensure normality and equality of variances followed by Wilcoxon signed-rank paired tests to compare the errors were conducted. In order to test whether the error increase was associated with clinical parameters, correlation analysis was performed. Statistical differences between clinical parameters at different ranges of error (<5%, 5–10%, >10%) were investigated as well. Correlation analyses were also carried out to explore the association between the estimated distance during 2MWT, performed either at hospital or at home, and clinical outcomes, including 2MWD, 6MWD, T25FW, EDSS, and FSS. Home-based estimations of the 2MWD during the week after each clinical visit were paired with the clinical outcomes obtained at the hospital. Pearson’s correlation was calculated with the significance threshold to be set at *p*-value < 0.01.

## 3. Results

Results of statistical analysis showed that the correlation coefficients for the personalized and fixed step lengths were r=0.55 and r=0.70, respectively. The null hypothesis that the two correlations are not significantly different was rejected (*p*< 0.001). Moreover, the absolute error using the fixed step lengths (15.33 ± 13.22 m) was statistically significantly (*p*< 0.001) higher than the error of personalized step lengths (11.23 ± 12.37 m). Although the error did not correlate with clinical parameters, statistically significant differences in disability severity scores (3.06±1.01 vs. 3.44±1.03) and step lengths (0.69±0.07 m vs. 0.66±0.09 m) were found between smaller (ε≤5%) and larger errors (ε>10%), respectively.

The performance of the two algorithms to automatically assess the 2MWT performed in clinical settings is summarized in Table 2 and Figure 2. Bland–Altman plots show that both *D* and $D_{i}$ tend to overestimate 2MWD for smaller values, with performance evaluation using *D* to exhibit a higher bias (Figure 2). Results in Table 2 show that the error ε between the estimated and the reference 2MWD is lower when considering an individual-specific step length ($D_{i}$) than a fixed one (*D*). For the distance $D_{i}$, a RAE below 10% was obtained for about two-thirds of 2MWT suggesting that the 2MWD can be measured robustly with the usage of wearable devices in clinical settings.

Table 3 illustrates the correlation values between the estimated 2MWDs and various clinical outcomes. Results show statistically significant strong correlation between walk test scores (2MWD, 6MWD, and T25FW) and the distance $D_{i}$, estimated either at hospital or at home. Larger correlation values (r>0.7) were obtained for the 2MWT performed at hospital than at home. Strong and moderate associations were found with EDSS and FSS scores, yielding r=−0.57 and r=−0.35, respectively. Regarding home-based estimation, smaller or similar correlation values were observed for disability and fatigue severity, respectively. Scatter plots between the estimated distance $D_{i}$, both at home and at hospital, and various clinical outcomes are shown in Figure 3.

## 4. Discussion

This study explores the feasibility of a consumer-friendly wearable device for automatic assessment of the 2MWD in pwMS with MS in clinical and non-clinical settings. Furthermore, the potential for remote step count monitoring to improve the evaluation of MS clinical outcomes is also investigated. Two step-based algorithms that take into account a fixed or an individual-specific step length are subjected to analysis. The accuracy of the algorithms is evaluated in clinical settings, and the correlation of the estimated 2MWD, obtained in and out of clinical settings, with different clinical outcomes is also studied.

Results from error analysis show that the step-based algorithm that includes individual-specific step lengths ($D_{i}$) performed better than the algorithm considering a fixed step length for all individuals (*D*) (Table 2). For instance, in 66.1% of the 2MWTs evaluated in clinical settings, the error ε in estimating the distance with $D_{i}$ was less than 10%, while *D* was equally accurate in 49.5% of the walk tests. Furthermore, using $D_{i}$, 43.9% of the 2MWDs exhibit an error ε of less than 5%. Previous studies found similar errors for 2MWTs performed at hospital, with 48.4% of pwMS exhibiting measurement errors of less than 5% [28]. Other studies examining the accuracy and precision of wearable motion sensors reported even lower errors, but their primary limitation is that the walk tests were conducted on a motorized treadmill [29,30]. Larger errors in estimating 2MWDs may be attributed to differences in disability severity (EDSS 3.06 ± 1.01 for ε≤5%, EDSS 3.44 ± 1.03 for ε>10%). The possibility that higher disability influenced the degree of accuracy in individuals with MS was previously documented in [31]. Considering that a clinically meaningful change in pwMS was found to be 10.2% ± 25.1% [32], an error between −30% and 30% for $D_{i}$ (Figure 2) may still be valid for the vast majority of patients. These findings imply that robust estimation of the 2MWD in clinical settings can be feasible in the majority of pwMS with the usage of wearable devices.

Results from correlation analysis show that there is a significant association between 2MWD and the distance $D_{i}$, not only estimated at hospital but also at home, yielding r=0.71 and r=0.58, respectively (Table 3). Similar performance between manual and remote step count monitoring during a single trial of 2MWT at hospital was reported in [33]. Lower correlation values obtained from home-based assessments compared to clinical settings can be attributed to numerous factors. The testing environment, such as the length of the corridors and the number and frequency of subject turns, can have a significant impact on how similar the 2MWT performance was in and out of clinical settings [25]. Furthermore, given that some MS-related symptoms—fatigue, pain, and mood in particular—vary dynamically across and within days, the time of the 2MWT execution may be an important consideration [34,35]. The provision of encouragement and the course description only at hospital may be also serve as possible moderator variables of walk test performance [36]. Although there are some factors that can affect the performance, it can be concluded that wearable devices can be used for automatic assessment of walk tests in pwMS in and out of clinical settings.

Wearable devices can also enhance the assessment of various MS-related clinical outcomes. In addition to the strong association with 2MWD, $D_{i}$ was also significantly correlated with 6MWD and T25FW for assessments both at hospital (r>0.7) and at home (r>0.5) (Table 3). Previous research has shown that there is a strong relation among walk test scores [14,37]. It should be noted that using a fixed step length worsened the association with walk test scores. Moderate-to-strong (r>0.3) correlations of $D_{i}$ with EDSS and FSS suggest that remote monitoring of disability and fatigue is feasible; however, there is still margin for improvement. For instance, in addition to the distance traveled, the measurement of spatiotemporal variables during walk tests, such as step and stride regularity, could provide a more detailed idea of individual performance [38]. Alterations in balance and the presence of falls in pwMS could provide additional information for evaluating ambulation [39]. Recent evidence indicates that balance significantly correlates with spasticity in pwMS, while the presence of spasticity symptoms in the lower limbs, including muscle spasms or stiffness, is associated with impairments in ambulation [40,41]. Spasticity may also predict future falls [39].

Objective gait-based biomarkers can assist with the identification of disability evolution in pwMS [42,43]. In addition to gait analyses, autonomic function can be assessed using wearable technology [44]. Emerging evidence suggests that autonomic response to walk tests can also be useful for assessing MS-related outcome measures [45]. In [46], pwMS who were able to adjust cardiac and ventilatory values were associated with better clinical outcomes. In [47], higher oxygen consumption (energy cost) during walking was related with slower cadence and shorter step length in MS. Increased oxygen cost was also associated with increased sensation of fatigue in pwMS due to higher metabolic heat and energy demand [48].

Despite the enormous scientific potential of wearable technology, some potential limitations need to be considered. In this study, longitudinal variations in MS symptoms or severity could not be assessed robustly since data for more than three clinical visits were available in a only small subset (about 5%) of participants. To achieve higher engagement, accelerometer signals acquired with more conventional wearable devices, e.g., activity trackers, smartphones [19], and actigraphy for unintentional walk testing during free-living activities, can be considered [49]. Although the number of steps is valuable information, an individual-specific rather than a fixed step length should be evaluated at least once in a clinical setting. However, such approach has not been tested yet for long-term follow-ups whereby step length may vary with disease evolution. Moreover, the algorithms developed in this study should be adapted to be able to assess movement over extended periods and during free-living activities.

## 5. Conclusions

According to the results of this study, automatic assessment of the 2MWD in pwMS is feasible with the usage of a consumer-friendly wearable device in clinical and non-clinical settings. In 66.1% of the 2MWTs evaluated in clinical settings, the error in estimating the distance was less than 10% and the correlation with the reference distance was r=0.71. Moderate-to-strong correlations between the home-based assessment of the 2MWD and walk test, disability, and fatigue severity scores suggest that wearable devices can enhance the assessment of MS-related clinical outcomes.

## Figures and Tables

**Figure 1 sensors-23-06017-f001:**
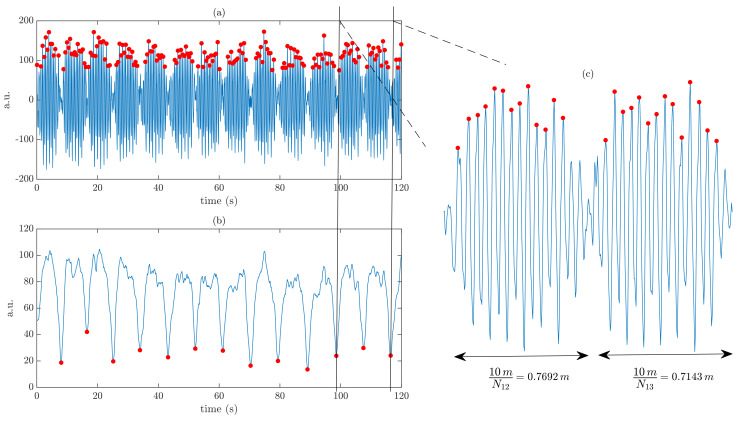
Examples of 2MWD estimation: (**a**) step detection (N=176), (**b**) turn detection, and (**c**) average step length estimation ($L_{i}=0.754$ m). The resulting estimated walking distances are D=127.42 m and $D_{i}=132.70$ m.

**Figure 2 sensors-23-06017-f002:**
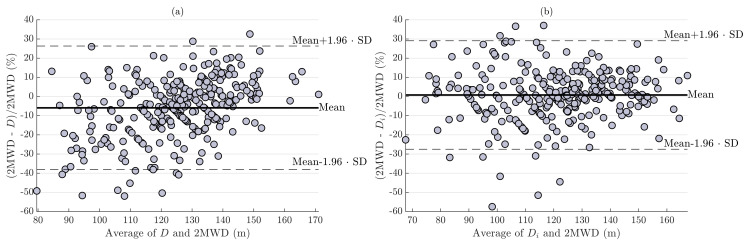
Normalized Bland–Altman plots for 2MWD estimation in clinical settings. Performance evaluation using (**a**) *D* and (**b**) $D_{i}$. Bias and limits of agreement are shown as solid and dotted black lines, respectively.

**Figure 3 sensors-23-06017-f003:**
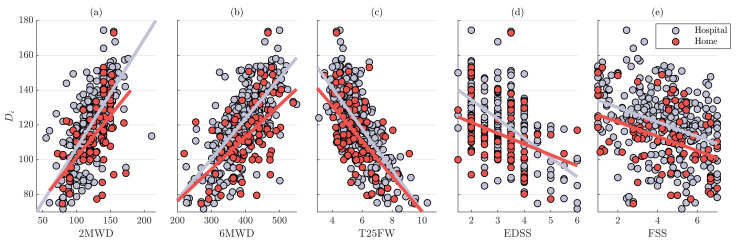
Scatter plots between clinical outcomes and the 2MWDs estimated using the individual-specific step length $D_{i}$. The clinical outcomes are (**a**) 2MWD, (**b**) 6MWD, (**c**) T25FW, (**d**) EDSS, and (**e**) FSS. A grey line and a red line are fitted to the data obtained from 2MWTs performed at hospital and home, respectively.

**Table 1 sensors-23-06017-t001:** Mean ± standard deviation of demographics and clinical outcomes assessed at hospital.

Number of participants	154
Phenotype (RRMS/SPMS)	130/24
MS duration, years	15.1 ± 8.9
Sex (female/male)	91/63
Age, years	47.3 ± 9.3
BMI, kg/m${}^{2}$	24.5 ± 4.9
Number of clinical visits, *n*	323
2MWD, m (*n*)	122.04 ± 22.82 (323)
6MWD, m (*n*)	387.23 ± 65.70 (317)
T25FW, s (*n*)	5.75 ± 1.24 (323)
EDSS (*n*)	3.2 ± 1.0 (239)
FSS (*n*)	4.4 ± 1.7 (285)

RRMS, Relapsing–Remitting Multiple Sclerosis; SPMS, Secondary Progressive Multiple Sclerosis; BMI, body mass index; 2MWD, 2-Minute Walk Distance; 6MWD, 6-Minute Walk Distance; T25FW, Timed 25-Foot Walk; EDSS, Expanded Disability Status Scale; FSS, Fatigue Severity Scale.

**Table 2 sensors-23-06017-t002:** Performance evaluation of algorithms for 2MWD estimation in clinical settings.

RAE	Hospital	
	*D*	$D_{i}$
ε≤5%	27.2%	43.9%
5%<ε≤10%	22.3%	22.6%
ε>10%	50.5%	33.5%

Abbreviations: 2MWD, 2-Minute Walk Distance; RAE, Relative Absolute Error; *D*, 2MWD estimated from accelerometer data considering a fixed step length for all individuals; $D_{i}$, 2MWD estimated from accelerometer data considering a different step length for each individual.

**Table 3 sensors-23-06017-t003:** Correlation analysis between clinical outcomes and the estimated 2MWD.

Outcomes	Hospital		Home	
	*D*	$D_{i}$	*D*	$D_{i}$
2MWD	$\;0.55{\;}^{*}$	$\;0.71{\;}^{*}$	$\;0.41{\;}^{*}$	$\;0.58{\;}^{*}$
6MWD	$\;0.53{\;}^{*}$	$\;0.72{\;}^{*}$	$\;0.27{\;}^{*}$	$\;0.57{\;}^{*}$
T25FW	$-0.52{\;}^{*}$	$-0.72{\;}^{*}$	$-0.34{\;}^{*}$	$-0.60{\;}^{*}$
EDSS	$-0.51{\;}^{*}$	$-0.57{\;}^{*}$	−0.21	$-0.33{\;}^{*}$
FSS	$-0.21{\;}^{*}$	$-0.35{\;}^{*}$	−0.22	$-0.38{\;}^{*}$

Statistically significant correlation values are marked with an asterisk: 2MWD, 2-Minute Walk Distance; 6MWD, 6-Minute Walk Distance; T25FW, Timed 25-Foot Walk; EDSS, Expanded Disability Status Scale; FSS, Fatigue Severity Scale; *D*, 2MWD estimated from accelerometer data considering a fixed step length for all individual; $D_{i}$, 2MWD estimated from accelerometer data considering a different step length for each individual.

## Data Availability

Data collected in the context of RADAR-CNS project are unavailable due to privacy.

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
