# Peer review of "Automatic Assessment of the 2-Minute Walk Distance for Remote Monitoring of People with Multiple Sclerosis"

_sensors, 2023, doi:10.3390/s23136017_

Round 1
Reviewer 1 Report
I would like to thank the editor for the opportunity to review the present manuscript, which highlights the use of a system to automatically the parameters related to the 2-minute walk distance test in people with Multiple Sclerosis. The opportunity to collect different data based on simple and cheap technology can be beneficial in the clinical setting and might help to evaluate the effects of different therapeutic approaches better.
The manuscript is well written and well presented, I have only few minor comments/suggestions I hope can be helpful:
- spasticity could be a feature of some people with MS. I could not find if this symptom was present in the sample included in this study, as it might have a role in walking biomechanics and therefore could be considered when applying movement-analysis systems and devices (Norbye, Midgard and Thrane, Physiother Res Int, 2020; Balantrapu et al., Mult Scler Int, 2014). Can you please indicate if it was an exclusion criterium?
- when discussing gait and fatigue, it might be worth mentioning that the energy cost of walking represents a fundamental factor, as it might depend on both biomechanics and physiological responses, and therefore influence the development of fatigue, both in pwMS with and without spasticity (Jeng et al., NeuroRehabilitation, 2018; Buoite Stella et al., Eur J Appl Physiol, 2020); also, it might be interesting to mention how walking on ground and walking on an artificial terrain as a treadmill is different in terms of biomechanics and energetics.
Author Response
Please find attached a point-by-point response to your comments.

Reviewer 2 Report
In this paper, the authors have tested a device equipped with accelerometers and have assessed people with MS both through standard clinical tests and at-home variations. The authors conclude that the at-home assessment correlates well with clinical scores and could be used as an alternative or add-on assessment.
The approach the authors follow is very valuable. My main ‘remark’ or suggestion is that data acquired through continuous accelerometers contains way more information than what is currently available in clinical practice. As a first step, a correlation with clinical standards may indicate feasibility. Still, I would expect that these datasets may teach us more about the patient than what we currently acquire.
One of the results seems to be that the personalized step length is a better way of estimating the 2MWD. I have two main issues with this result. First, I am wondering why the authors assumed that a fixed step length for all subjects could work. I would assume that step length would, e.g., be strongly affected by physical disability. Further, the statistical analysis does not support this claim as I cannot find a direct comparison between the obtained correlation coefficients (r1: Di vs 2MWD and r2: D vs 2MWD). No statistical comparison was included in table 2 either. Please further substantiate this claim.
- the 2MWD is based on an estimation of the participants’ step length, but wouldn’t this step length also vary with disease evolution? How would you use such a technology to follow-up MS patients as the technology probably becomes less reliable the longer the follow-up?
- the instructions for the 2MWD are such that the patients should walk along a 10 m-long corridor at the maximum possible speed. How many patients reported having a 10m long corridor at home, and how did the researchers handle the risk of the patient falling?
- how does the wearable device look like and how does it compare to the accelerameters included in smartphone or smartwatch devices? Please include a brief description inside this paper and do not just refer to another one.
- would the device be able to function properly in the case of spasticity?
- I do not understand how the researchers arrived at L = 0.724m. Given the importance of the paper, I think a brief recap of the paper to which the authors refers may be appropriate.
- When considering Figure 2, about 95% of the observations have an error between -40m and +30m (for D) and -30 and +30 for Di; how should I interpret this? Is this clinically useful? How small should the error be to become clinically useful? What is a meaningful clinical change in the 2MWD? I would assume that an error of 30-40m on the 2MWD that has a spread between 75 and 180m is simply too large to be of any use?
- Wouldn’t you expect that the outcome of the FSS would vary quite substantially during the day and that a single walking test would not consitute a substitute for a questionnaire or a VAS scale that could be incorporated in an app?
- the authors claim that this study showed that “although, the number of steps has valuable information, an individual, rather than a fixed step length should be evaluated at least once at clinical settings”. I actually do not think that the authors have tested this statistically. I don’t see a comparison of the correlation coefficients between D-2MWD or Di-2MWD. I also don’t see a comparison of the error estimated with both methods.
- does the error increase with increasing disability? Ie does the accuracy depend on the level of disability? are there any clinical differences between the groups described in Table 2? I now see in the discussion that this is indeed the case (this should be in the results). This should be more elaborated. I am also wondering why the authors artificially split their group of participants in discrete cohorts - why not run a correlation analysis? And why only focussing on the EDSS and not compare the other clinical parameters? Does the error increase also corrrelate with an individual’s step length?
- How does the walking speed evolve during the 2MWD? There is a lot of novelty waiting here that could be relatively easily included
Author Response

(The authors gave the same response as above.)

Round 2
Reviewer 2 Report
Dear authors,
Many thanks for the additional analyses and clarifications. Congratulations on your work!